# Persistence of symptoms and quality of life at 35 days after hospitalization for COVID-19 infection

**Laurie G. Jacobs**[1,2◉]*, **Elli Gourna Paleoudis**[3◉], **Dineen Lesky-Di Bari**[4‡],
**Themba Nyirenda**[3‡], **Tamara Friedman**[3], **Anjali Gupta**[1,2], **Lily Rasouli**[2],
**Marygrace Zetkulic**[1,2], **Bindu Balani**[1,5], **Chinwe Ogedegbe**[6,7], **Harinder Bawa**[1,2,8,9],
**Lauren Berrol**[10], **Nabiha Qureshi**[2], **Judy L. Aschner**[8,9]

1 Department of Internal Medicine, Hackensack Meridian School of Medicine, Nutley, New Jersey, United States of America, 2 Department of Internal Medicine, Hackensack University Medical Center, Hackensack, New Jersey, United States of America, 3 Office of Research Administration, Hackensack Meridian Health, Edison, New Jersey, United States of America, 4 Department of Research, Joseph M. Sanzari Children's Hospital at Hackensack University Medical Center, Hackensack, New Jersey, United States of America, 5 Division of Infectious Disease, Department of Internal Medicine, Hackensack University Medical Center, Hackensack, New Jersey, United States of America, 6 Department of Emergency Medicine, Hackensack Meridian School of Medicine, Nutley, New Jersey, United States of America, 7 Hackensack University Medical Center, Hackensack, New Jersey, United States of America, 8 Department of Pediatrics, Hackensack Meridian School of Medicine, Nutley, New Jersey, United States of America, 9 Joseph M. Sanzari Children's Hospital at Hackensack University Medical Center, Hackensack, New Jersey, United States of America, 10 Department of Anesthesiology, Hackensack University Medical Center, Hackensack, New Jersey, United States of America

◉ These authors contributed equally to this work.
‡ These authors also contributed equally to this work.
* Laurie.Jacobs@HackensackMeridian.org

## Abstract

### Background

Characterizing the prevalence and persistence of symptoms associated with COVID-19 infection following hospitalization and their impact is essential to planning post-acute community-based clinical services. This study seeks to identify persistent COVID-19 symptoms in patients 35 days post-hospitalization and their impact on quality of life, health, physical, mental, and psychosocial function.

### Methods and findings

This prospective cohort study used the PROMIS® Instruments to identify symptoms and quality of life parameters in consecutively enrolled patients between March 22 and April 16, 2020, in New Jersey. The 183 patients (median age 57 years; 61.5% male, 54.1% white) reported persistent symptoms at 35 days, including fatigue (55.0%), dyspnea (45.3%), muscular pain (51%), associated with a lower odds rating general health (41.5%, OR 0.093 [95% CI: 0.026, 0.329], p = 0.0002), quality of life (39.8%; OR 0.116 [95% CI: 0.038, 0.364], p = 0.0002), physical health (38.7%, OR 0.055 [95% CI: 0.016, 0.193], p <0.0001), mental health (43.7%, OR 0.093 [95% CI: 0.021, 0.418], p = 0.0019) and social active role (38.7%,

**Data Availability Statement:** All relevant data are within the manuscript and its Supporting Information files.

**Funding:** The Authors received no specific funding for this work.

**Competing interests:** Laurie G. Jacobs -no conflicts of interest, no financial relationships Elli Gourna Paleoudis -no conflicts of interest, no financial relationships Dineen Lesky-DiBari - no conflicts of interest, no financial relationships Anjali Gupta -no conflicts of interest, no financial relationships Lily Rasoulli - no conflicts of interest, no financial relationships Themba Nyrirenda - no conflicts of interest, no financial relationships Tamara Friedman - no conflicts of interest, no financial relationships Marygrace Zetkulic -no conflicts of interest, no financial relationships Bindu Balani - Financial relationships: Gilead- PI, Remdisivir clinical trial, 2020, personal fees; Merck advisory committee; Abbvie speaker, personal fees Chinwe Ogedegbe -no conflicts of interest, no financial relationships Harinder Bawa -no conflicts of interest, no financial relationships Nabiha Qureshi -no conflicts of interest, no financial relationships Lauren Berrol -no conflicts of interest, no financial relationships Judy Aschner -Gilead Science, stock owner" We confirm that the competing interests and financial relationships statement above is correct and that this does not alter our adherence to PLOS ONE policies on sharing data and materials.

OR 0.095 [95% CI: 0.031, 0.291], p<0.0001), as very good/excellent, particularly adults aged 65 to 75 years (OR 8·666 [95% CI: 2·216, 33.884], p = 0·0019).

## Conclusions

COVID-19 symptoms commonly persist to 35 days, impacting quality of life, health, physical and mental function. Early post-acute evaluation of symptoms and their impact on function is necessary to plan community-based services.

## Introduction

In December 2019, a new type of coronavirus, now known as COVID-19 (SARS-CoV-2), was identified in China [1]. Within a few months of this report, COVID-19 has become a world-wide pandemic, posing a significant threat to public health. By mid-June 2020, there were over 10 million cases world-wide [2] with 2 million cases, including nearly 126,000 deaths, in the United States (US) [3].

Despite the vast numbers of individuals who have been infected and suffered from COVID-19, the clinical course of the associated respiratory illness, including prevalence, persistence of signs and symptoms, and the impact on general wellbeing and function, has not been fully described. Initial signs and symptoms leading to hospitalization include fever, cough, dyspnea, tachypnea, fatigue and reduced oxygen saturation requiring supplemental oxygen [1, 3, 4]. Many patients recover sufficiently to be discharged from hospital within seven to ten days but may still not be free of symptoms. Others progress to a hyperinflammatory state and to adult respiratory distress syndrome (ARDS) [1, 5] which is associated with a high mortality attributed to either respiratory illness or heart failure [6, 7].

COVID-19 infection has other manifestations beyond the respiratory illness, including cardiac injury (cardiomyopathy, ventricular arrhythmias, hemodynamic instability in the absence of obstructive coronary artery disease) [8], thrombotic complications (including stroke, myocardial infarction, venous thromboses) [9], and renal, gastrointestinal, neurologic, among others [5]. Signs and symptoms of these conditions may overlap with those of the respiratory illness and contribute to the symptom burden of COVID-19 infection. The course and outcomes of these manifestations are mostly unknown.

Areas of uncertainty regarding community-dwelling patients early during the post-acute period include infectivity and the need for quarantine, symptom burden and persistence, evaluation and management of enduring symptoms, the need for surveillance for late manifestations, such as thrombotic complications, and the need for physical and psychological support. One study which informs the discussion of post-acute quarantine evaluated six individuals and found that after resolution of signs and symptoms, SARS-CoV-2 RNA was detectable for a median (range) of 26 days (9 to 48), and viral detection persisted for a median of 34 days (22 to 67) after hospitalization [10]. Without knowledge of infectivity and immunity, these data are relevant to planning for post-acute care and isolation.

The mid- and long-term effect of COVID-19 infections on patients' general health, wellbeing, physical function, and their ability to return to work, has not been elucidated. A recent communication from Italy evaluating 143 patients with COVID-19 infection [11], indicates persistence of symptoms and a reduced quality of life recalled at a 60-day visit after symptom onset.

Our study examines the early post-acute clinical trajectory of symptoms to 35 ± 5 days after hospital discharge for COVID-19 infection, their association with individual's self-rated quality of life, physical function, general health, mental health, emotional health, social relationships, active roles, and their ability to perform activities of daily living, in an early US pandemic community-dwelling cohort. Early post-acute evaluation is necessary to define ongoing clinical care.

## Materials and methods

### Study design and participants

This research was approved by the Hackensack University Medical Center IRB Pro2020-0440 for human subject research protection. This single center prospective cohort study of individuals with COVID-19 infection requiring hospitalization sought to identify as primary outcomes, the persistence of symptoms in patients with COVID-19 infection, specifically fatigue, shortness of breath (dyspnea), cough, lack of taste (dysgeusia), muscular pain, diarrhea, production of phlegm, headache, joint pain, eye irritation, fever, and confusion, and their impact on quality of life, general health, physical health, mental health, social relationships, social active roles and daily physical activities from hospital discharge to 35 (± 5) days later. Secondary outcomes included the relationship of participant characteristics, comorbid conditions, and oxygen requirements during and after hospitalization, with the persistence of symptoms and quality of life estimations at 35 (± 5) days after discharge and their impact on quality of life. This study was undertaken at an academic medical center in New Jersey, an early US epicenter of COVID-19 infection.

A survey comprised of 23–43 questions (depending upon responses), was administered by email or telephone call to participants 35 (± 5) days following their hospital discharge for COVID-19 infection. The survey asked participants to address: a) pre-infection general health status, living situation, and symptoms; b) discharge (baseline) general health status during the previous seven to fourteen days based upon the PROMIS® Scale v1.2 –Global Health [12]; c) daily activities within the past seven to fourteen days based on the PROMIS Item Bank v1.0 – Dyspnea Functional Limitations–Short Form 10a [13]; and d) the trajectory of symptoms after discharge, including timing of improvement. A focused review of the medical record for this admission was conducted to identify demographic data, comorbidities, oxygen requirements, length of hospital stay, and post-hospital clinical care.

Participants' severity of COVID-19-associated illness was scored using an ordinal scale developed by a special World Health Organization (WHO) committee [14] for use in randomized multi-center adaptive clinical trials and is based upon the site of care (community or hospital) and intensity of oxygen supplementation required. Scores range from zero (no clinical or virologic evidence of infection) to eight (death), with hospitalized patients characterized as having "mild disease" with scores of three (no oxygen therapy) or four (oxygen by mask or nasal prongs), and "severe disease" for patients with scores of five (non-invasive ventilation or high-flow oxygen), six (intubation and mechanical ventilation), or seven (ventilation plus additional organ support). No descriptions of symptoms are included.

Eligible participants were contacted via email (if available) or by telephone call, at 35 (± 5) days after hospital discharge. An initial email was sent, followed by two reminders on two subsequent weekdays, if no response was obtained. The email included a link providing access to the study description and an e-consent form. If the participant agreed to consent, this could be completed electronically, followed by the survey, using REDCap (Research Electronic Data Capture), a secure, web-based software platform that supports data capture [15, 16].

Participants without a valid email address or who did not respond by email were contacted at the phone numbers recorded in their medical record. If no response was obtained, they were called once daily, for a total of three attempts, on subsequent weekdays. As per local IRB standard operating procedures, no voice mail was left, no return calls were requested, and no proxies were used for the survey. The study was described over the telephone and consent was obtained using an IRB-approved script. If consent was obtained, participants completed the survey over the phone and their responses were logged by study staff directly into REDCap.

Eligible participants were identified from a list of consecutive hospital discharges between March 22 and April 16, 2020, after treatment for COVID-19 infection who met the following criteria: a) aged 18 or older; b) diagnosis of viral RNA PCR-confirmed COVID-19 infection during hospitalization; c) hospitalization duration of at least three days. Exclusion criteria included: a) individuals who had expired during or after hospitalization; b) non-English speakers (as recorded in their medical record) as the email was in English and certified translators redeployed to clinical duty were unavailable for telephone calls; and, c) individuals with a documented diagnosis of dementia or delirium (or otherwise unable to provide consent). For patients who were readmitted during the study period, only the most recent hospitalization was evaluated.

## Statistical analysis

Descriptive analysis was performed with continuous variables summarized by mean (standard deviation) or median (interquartile range), depending on whether the data were normally distributed. The assumption of normality was assessed using Shapiro-Wilk test of normality. Categorical variables were summarized as frequencies (percentage). Comparison of continuous variables between two groups were conducted using two-sided t-test or Wilcoxon rank sum test, as appropriate. Categorical variables were compared between groups using two-sided Fisher's exact test or Pearson's chi-square test, as appropriate. Unless specified otherwise, any p-value < 0.05 was considered statistically significant. All data analysis was performed using R software version 3.6.3 (The R Foundation for Statistical Computing, Vienna, Austria) and SAS version 9.4 (SAS Institute Inc., Cary, North Carolina, USA).

The association between any participant-reported persistent symptoms by day 35 (+/-5) post-discharge and participant characteristics, was examined by modelling the composite endpoint, binary outcome for none versus at least one persistent symptom, and for each symptom separately. For each of these binary outcomes, a univariate logistic regression analysis was conducted and an odds ratio, corresponding 95% confidence intervals, and p-values, with significance between 0.05 and 0.15, were tabulated for all variables. The association between quality of life outcomes and a composite endpoint of one or more persistent symptoms at 35 days was analyzed using an ordinal logistic regression analysis of the quality of life outcomes ordered response category versus one or more persistent symptoms. General health, with three ordered category levels "poor/fair," "good," and "very good/excellent," was modeled against the potential predictor. For items 1–6 of the PROMIS tool, variables were reclassified using the ordered three categories described for general health [17]. Ratings of physical activity were reclassified into ordered categories of "not at all, a little", "moderately" and "mostly, completely;" likewise, emotional was reclassified into "always, often", "sometimes" and "rarely, never, and fatigue, to "severe, very severe," "moderately" and "none, mild."

Logistic regression analysis of ordinal categorical outcomes, such as the nine quality of life outcomes, relies on the assumption of proportional odds being satisfied. The assumption is that odds of observing one level, versus all cumulative levels, are identical for any selected level and levels above. If this assumption is violated, such as in a three-category response, then two

separate logistic regressions are required to examine the association with potential predictor. Consequently, a generalized logit regression model which assumed unequal slopes and "poor/fair" as the reference response category was fit to examine the odds of "very good/excellent" health versus the odds of "poor/fair" health; and odds of "good" health versus odds of "poor/fair" health. Significant associations between having any persistent symptoms and the odds of a "very good/excellent" response regarding general health, physical health, mental health, social relationships, social active roles, activities of daily living, and discharge with supplemental oxygen, are reported.

## Results

A total of 183 participants enrolled and completed the survey, 23% by email (49 of 212 participants who received emails completed the survey), and 48% by telephone call (134 of the 279 participants who were called completed the survey), with an overall response rate of 52% (183 of 351 eligible).

The participants' median age was 57 years (interquartile range [IQR] 48–68; range 25–85), with fairly consistent representation across the decades, except at the extremes of age (Table 1). Males represented 61.5%. The median BMI (Body Mass Index) [18], was 30.0 (IQR 27.3–33.5), with 36% categorized as overweight and 49.1% as obese. Obesity, hypertension (47.5%), diabetes (28.4%), hyperlipidemia (10.9%), were highly prevalent, components of the metabolic syndrome [19]. The median length of hospital stay was seven days (IQR 5–10), with a range of three to 33 days.

The majority of participants (89.6% overall; 87.4% of males, 92.7% of females) required oxygen support during hospitalization (Table 2). The WHO severity criteria [14], identified that 95.4% of all participants, 91.9% of males, and 81.1% of females, were classified as having a mild severity of illness, which includes those requiring no oxygen, or oxygen by nasal prongs or nonrebreather mask. Supplemental oxygen was required at discharge as home therapy for 37 (20.3%) participants who met the US criteria based upon blood gas results [20].

The most frequent symptoms occurring at discharge recalled by the 183 participants were fatigue (104, 56.8%), shortness of breath (94, 51.4%), cough (74, 40.4%), lack of taste (40, 21.9%), muscular pain (37, 20.2%), diarrhea (29, 15.8%), lack of smell (28, 15.3%), production of phlegm (25, 13.7%), and headache (24, 13.1%). At the time of the survey (35 ± 5 days post-discharge), participants were asked to describe current symptoms and to recall symptoms which had stopped during the period of the first two weeks, the third, and the fourth week, after discharge. At 35 days, 82/149 (55.0%) reported fatigue, 37/77 (50.6%) muscular pain, 58/128 (45.3%) shortness of breath and 46/110 (41.82%) cough (Table 3). Older participants aged 65 to 75 years (OR 8.666 [95% CI: 2.216, 33.884], p = 0.0019), and women (male versus female gender [OR 0.462 {0.225, 0.949}, p = 0.0356]), had statistically significant higher odds of experiencing persistent symptoms. Persistent symptoms were reported by 133 participants (72.7%, [95% CI: 65.6, 78.9]) at day 35.

Comparing patients' baseline general health (pre-COVID-19 illness) to post-COVID-19 general health (at time of survey) indicated that patients who reported excellent/very good or good health post-COVID-19 were more likely to have reported an excellent/very good or good pre-COVID health status (Table 4). Similarly, reporting an excellent/very good quality of life, physical health and mental health post-COVID-19 was more frequent among those reporting excellent/very good pre-COVID health.

At the time of the survey, a minority of participants reported experiencing "very good/excellent" quality-of-life, physical and mental health, social relationships, and participation in active roles (Table 5). Self-rated ability to perform activities of daily living declined with

**Table 1. Characteristics of discharged patients with COVID-19 infection.**

| Demographics/comorbid conditions[a,b,c] | | no. (%) |
|---|---|---|
| Age median (IQR), years | 57 (48–68) | |
| Age range, years | 25–85 | |
| Age decile (years) | 25–35 | 14 (7.6) |
| | 35–45 | 22 (12.0) |
| | 45–55 | 42 (23.0) |
| | 55–65 | 46 (25.1) |
| | 65–75 | 45 (24.6) |
| | 75–85 | 14 (7.7) |
| Sex | Male | 112 (61.5) |
| Race | White | 99 (54.1) |
| | Black or African American | 16 (8.7) |
| | Asian | 16 (8.7) |
| | American Indian/Alaskan | 2 (1.1) |
| | Other, non-white | 50 (27.3) |
| Ethnicity | Hispanic | 55 (30.2) |
| | Non-Hispanic | 120 (65.9) |
| | Unknown | 7 (3.8) |
| BMI,[c] median (IQR), Kg/m$^2$ | | 30.0 (27.3–33.5) |
| Obesity class[d] | Overweight (BMI 25.0–29.9) | 66 (36.1) |
| | Obese (BMI $\geq$ 30.0) | 90 (49.2) |
| Hypertension | | 87 (47.5) |
| Diabetes | | 52 (28.4) |
| Cardiac disease | CAD or history of MI | 21 (11.5) |
| | Arrythmia | 9 (4.9) |
| | Heart failure | 5 (2.7) |
| | Other | 16 (8.7) |
| Hyperlipidemia | | 20 (10.9) |
| Asthma | | 19 (10.4) |
| Cancer | | 18 (9.8) |
| Immunodeficiency[e] | | 8 (4.4) |
| Hypothyroidism | | 8 (4.4) |
| Psychiatric disorders[f] | | 8 (4.4) |
| Obstructive sleep apnea | | 6 (3.3) |
| COPD | | 7 (3.8) |
| Thromboembolic disease | | 3 (1.6) |

Footnotes

[a]Abbreviations: IQR Interquartile range; BMI, body mass index (calculated as weight in kilograms divided by height in meters squared), SD standard deviation, CAD coronary artery disease, MI myocardial infarction, COPD chronic obstructive pulmonary disease.

[b]Demographic information obtained from self-reported hospital registration record.

[c] Co-morbid conditions identified from the medical record at admission.

[d] Obesity as defined by CDC [19].

[e] Immunodeficiency is secondary to co-morbid HIV (1), rheumatoid arthritis on disease-modifying agents or steroids (5), or renal transplant on immunosuppressive therapy (2).

[f] Psychiatric disorders included anxiety, depression, schizophrenia.

**Table 2. Oxygen requirements during hospitalization for COVID-19.**

| | **All Participants** | **Male Sex[a]** | **Female Sex[a]** | **WHO Severity [15]** |
|---|---|---|---|---|
| **Oxygen Delivery Requirement** | *Count (% of 183)* | *Count (% of 111)* | *Count (% of 69)* | |
| None | 19 (10·4) | 14 (12·6) | 5 (7·2) | Mild |
| Nasal prongs | 118 (72·4) | 73 (65·8) | 43 (62·3) | Mild |
| Nonrebreather mask | 23 (12·6) | 15 (13·5) | 8 (11·6) | Mild |
| High-Flow | 15 (9·2) | 6 (5·4) | 8 (11·6) | Severe |
| Intubation and mechanical ventilation | 8 (4·94) | 3 (2·7) | 5 (7·2) | Severe |

Footnotes:

[a] Sex: 111 male, 69 female, 3 unknown.

increased physical effort, such as climbing stairs, lifting and carrying, and walking fast (Table 5).

For all of the PROMIS tool items, except social relationships, the test of the validity of proportional odds assumption in the ordinal response categories of the quality of life was not supported by the data (p-values ranging from < .0001 to 0.0047). Social relationships responses only consisted of "poor/fair" outcomes and "very good/excellent" outcomes. Consequently, for each the ordinal QOL outcomes, two logistic regression analyses were reported to reflect the unequal slopes.

The most prevalent comorbidities—hypertension, obesity, diabetes and cardiovascular disease—did not appear to have an effect on post-COVID-19 quality of life responses (captured at the time of the survey and during the prior seven days). Similarly, these co-morbidities did not affect post-COVID-19 mental health, social relationships and general health responses. Prior

**Table 3. Participant-reported symptoms persisting from hospital discharge to 35 days in COVID-19 patients.**

| Symptom[a] | Ever Experienced, no. (% of total) | 2 weeks,[b] no./ever experienced (%) | 3 weeks, no./ever experienced (%) | 35 ± 5 days,[c] no./ever experienced (%) |
|---|---|---|---|---|
| Fatigue | 149 (83.2) | 120/149 (80.5) | 91/148 (61.1) | 82/149 (55.0) |
| Shortness of breath | 128 (71.1) | 94/128 (73.3) | 69/127 (54.0) | 58/128 (45.3) |
| Cough | 110 (61.4) | 78/110 (70.9) | 55/110 (50.0) | 46/110 (41.8) |
| Lack of taste | 79 (44.4) | 45/79 (57.0) | 28/79 (35.4) | 18/79 (22.8) |
| Muscular pain | 77 (43.0) | 57/77 (74.0) | 41/77 (53.2) | 39/77 (50.6) |
| Diarrhea | 64 (36.0) | 32/64 (50.0) | 13/64 (20.3) | 7/64 (10.9) |
| Lack of smell | 65 (36.7) | 39/65 (60.0) | 25/65 (38.5) | 17/65 (26.2) |
| Phlegm | 61 (34.7) | 46/61 (75) | 31/61 (50.8) | 27/61 (44.3) |
| Headache | 59 (33.2) | 42/59 (71.2) | 29/59 (49.2) | 23/59 (39.0) |
| Joint pain | 53 (29.8) | 46/53 (86.8) | 29/53 (54.7) | 29/53 (54.7) |
| Confusion | 37 (21.1) | 25/37 (67.6) | 19/37 (51.4) | 16/37 (43.2) |
| Eye irritation | 35 (19.8) | 20/35 (57.1) | 18/35 (51.4) | 15/35 (42.9) |
| Fever | 38 (22.0) | 21/38 (55.3) | 5/38 (13.2) | 2/38 (5.3) |
| Ulcer | 10 (5.7) | 5/10 (50.0) | 3/10 (30.0) | 2/10 (20.0) |

Footnotes

[a] Report of symptoms using PROMIS® survey questions, [12, 13] retrospective except at the time of survey regarding the period of time described.

[b] % = (number of participants reporting symptoms which had not stopped within indicated time period *100)/ (total number of participants ever reporting the symptom).

[c] % = (number of participants reporting symptoms present at the time of survey*100/total number of participants ever reporting the symptom).

**Table 4. Associations between pre-COVID-19 general health and post-COVID-19 (35 days) general health and quality of life.**

| Outcome | Effect | Odds Ratio | 95% CI | P-Value |
|---|---|---|---|---|
| | (reference. category: Poor, Fair) | | | |
| General Health<br>Very Good, excellent | pre-COVID-19 General Health | 5.248 | (1.093, 25.20) | 0.0384 |
| General Health<br>Good | Good | 4.017 | (1.130, 14.28) | 0.0317 |
| General Health<br>Very Good, excellent | pre-COVID-19 General Health<br>Very good, excellent | 8.847 | (2.178, 35.93) | 0.0023 |
| General Health<br>Good | pre-COVID-19 General Health<br>Very good, excellent | 2.442 | (0.792, 7.533) | 0.1203 |
| Quality of Life<br>Very Good, excellent | pre-COVID-19 General Health<br>Good | 2.916 | (0.618, 13.75) | 0.1762 |
| Quality of Life<br>Good | pre-COVID-19 General Health<br>Good | 1.296 | (0.383, 4.390) | 0.6767 |
| Quality of Life<br>Very Good, excellent | pre-COVID-19 General Health<br>Very good, excellent | 5.478 | (1.301, 23.07) | 0.0204 |
| Quality of Life<br>Good | pre-COVID-19 General Health<br>Very good, excellent | 1.285 | (0.421, 3.920) | 0.6594 |
| Physical Health<br>Very Good, excellent | pre-COVID-19 General Health<br>Good | 6.500 | (1.199, 35.23) | 0.0300 |
| Physical Health<br>Good | pre-COVID-19 General Health<br>Good | 5.500 | (1.451, 20.85) | 0.0122 |
| Physical Health<br>Very Good, excellent | pre-COVID-19 General Health<br>Very good, excellent | 13.20 | (2.747,63.44) | 0.0013 |
| Physical Health<br>Good | pre-COVID-19 General Health<br>Very good, excellent | 4.320 | (1.249, 14.95) | 0.0209 |
| Social Relationship | pre-COVID-19 General Health<br>Good | 1.587 | (0.487, 5.173) | 0.4439 |
| Social Relationship | pre-COVID-19 General Health<br>Very good, excellent | 2.121 | (0.717, 6.279) | 0.1744 |

comorbidities, however, did affect post-COVID-19 physical health responses. Patients with hypertension and diabetes, in particular, were less likely to report good or very good/excellent physical health. Those with hypertension trended to be less likely to have a physical health response of good (OR 0.428, [95% CI: 0.199, 0.921], p-value = 0.0300), and diabetics were less likely to have physical health responses of good (OR 0.330, [95% CI: 0.145, 0.747], p-value = 0.0079) and very good/excellent (OR 0.308, [95% CI: 0.138, 0.686], p-value = 0.0040). In addition, responses of excellent, very good and good physical health were more likely to be reported among those who reported having no known comorbidities at admission (OR = 2.822, p-value: 0.05 and OR = 3.600, p-value 0.01 for excellent/very good health and good health respectively).

At least one or more persistent symptoms at 35 days after hospital discharge were associated with reduced odds of participants reporting the most favorable outcome, such as "very good to excellent" responses for quality of life, general health, physical health, mental health, social relationships and active roles, "mild to none" for level of fatigue and "rarely to never" for emotion (Table 6). Reports of having experienced any persistent symptoms at day 35 was associated with a 94.5% reduced chance of reporting "very good to excellent" physical health (OR 0.055 [96% CI: 0.016, 0.193], p<0.0001). In the next lower quality of life rating comparisons

**Table 5. COVID-19 patient self-rated quality of life and activities of daily living at day 35.**

| Quality of life and daily living activity outcomes[a] | Rating | no. (%) |
|---|---|---|
| General health | poor, fair | 37 (20.2) |
| | good | 70 (38.2) |
| | very good, excellent | 76 (41.5) |
| Quality of life | poor, fair | 42 (23.2) |
| | good | 67 (37.0) |
| | very good, excellent | 72 (39.8) |
| Physical health | poor, fair | 49 (27.1) |
| | good | 62 (34.2) |
| | very good, excellent | 70 (38.7) |
| Mental health | poor, fair | 31 (16.9) |
| | good | 72 (39.3) |
| | very good, excellent | 80 (43.7) |
| Social Relationships | poor, fair | 110 (60.4) |
| | very good, excellent | 72 (39.6) |
| Social Active Role | poor, fair | 57 (31.5) |
| | good | 54 (29.8) |
| | very good, excellent | 70 (38.7) |
| Physical activity | not at all, a little | 25 (13.8) |
| | moderately | 45 (24.9) |
| | mostly, completely | 111 (61.3) |
| Emotional | always, often | 26 (14.2) |
| | sometimes | 64 (35.0) |
| | rarely, never | 93 (50.8) |
| Fatigue | severe, very severe | 15 (8.2) |
| | moderately | 60 (32.8) |
| | none, mild | 108 (59.0) |
| Dressing | no or a little difficulty | 177 (96.7) |
| | some/much difficulty | 6 (3.3) |
| Walking | no or a little difficulty | 146 (84.4) |
| | some/much difficulty | 27 (15.6) |
| Stairs | no or a little difficulty | 115 (70.1) |
| | some/much difficulty | 49 (29.9) |
| Meal preparation | no or a little difficulty | 148 (93.7) |
| | some/much difficulty | 10 (6.3) |
| Wash dishes | no or a little difficulty | 153 (95.6) |
| | some/much difficulty | 7 (4.4) |
| Sweep | no or a little difficulty | 133 (88.1) |
| | some/much difficulty | 18 (11.9) |
| Make bed | no or a little difficulty | 150 (93.8) |
| | some/much difficulty | 10 (6.2) |
| Lift | no or a little difficulty | 129 (79.6) |
| | some/much difficulty | 33 (20.4) |
| Lift and carry | no or a little difficulty | 120 (74.5) |
| | some/much difficulty | 41 (25.5) |

(*Continued*)

**Table 5.** (Continued)

| Quality of life and daily living activity outcomes[a] | Rating | no. (%) |
|---|---|---|
| Walk fast | no or a little difficulty | 74 (54.4) |
| | some/much difficulty | 62 (45.6) |

Footnotes

[a] Report of quality of life and daily living activities using PROMIS® survey questions, [12, 13] retrospective except at the time of survey.

"good" to "poor or fair," none of the quality of life outcomes were significantly associated with experiencing persistent symptoms by day 35. A rating of "good" as compared to "poor or fair" for any of the quality of life outcomes, was not statistically associated with persistent symptoms.

Participants experiencing persistent symptoms had a lower likelihood of being able to "mostly/completely" perform activities of daily living (OR 0.059 [95% CI: 0.008, 0.451], p = 0.0064). Specifically, experiencing one or more persistent symptom was associated with a six to seven fold increased odds of describing "some/much difficulty" with walking (OR 6.122 [95% CI: 1.392, 26.923], p = 0.0165), climbing stairs (OR 6.719, [95% CI, 2.259, 19.986], p = 0.0006), lifting (OR 7 .749 [95% CI: 1.771, 33.905], p = 0.0065), and walking fast (OR 7.110 [95% CI: 2 .724, 18.555], p<0.0001).

The reduced likelihood of rating physical health "very good/excellent" as compared to "poor/fair," was significantly associated with reported persistence of shortness of breath (OR 0.133 [95% CI: 0.043, 0.410], p = 0.0004), cough (OR 0.097 [95% CI: 0.028, 0.337], p = 0.0002), fatigue (OR 0.128 [95% CI: 0.052, 0.316], p<0.0001), or muscular pain (OR 0.161 [95% CI: 0.0478, 0.557], p = 0.0039). Participants were marginally less likely (p = 0.0515) to report

**Table 6.** Associations between quality of life outcomes and persistent symptoms of COVID-19 patients at day 35.

| Quality of life outcome variable | Outcome comparison category | Odds ratio (95% CI) | P-value |
|---|---|---|---|
| General health | good | 0.530 (0.136, 2.057) | 0.3587 |
| (ref. cat.: poor, fair)[a] | very good, excellent | 0.093 (0.026, 0.329) | 0.0002 |
| Quality of life | good | 0.536 (0.159, 1.809) | 0.3149 |
| (ref. cat.: poor, fair) | very good, excellent | 0.116 (0.038, 0.364) | 0.0002 |
| Physical health | good | 0.384 (0.098, 1.504) | 0.1694 |
| (ref. cat.: poor, fair) | very good, excellent | 0.055 (0.016, 0.193) | <0.0001 |
| Mental health | good | 0.286 (0.061, 1.342) | 0.1125 |
| (ref. cat.: poor, fair) | | 0.093 (0.021, 0.418) | 0.0019 |
| Social relationships | very good, excellent | 0.245 (0.123, 0.486) | |
| (ref. cat.: poor, fair) | | | |
| Social active role | good | 0.196 (0.060, 0.637) | |
| (ref. cat.: poor, fair) | very good, excellent | 0.095 (0.031, 0.291) | |
| Daily physical activity | moderately | 0.584 (0.058, 5.925) | |
| (ref. cat.: a little not at all) | mostly, completely | 0.059 (0.008, 0.451) | |
| Emotional | sometimes | 0.876 (0.252, 3.051) | |
| (ref. cat: always, often) | rarely, never | 0.301 (0.096, 0.947) | |
| Fatigue | moderately | 0.786 (0.085, 7.276) | |
| (ref. cat.: severe, very severe) | none, mild | 0.104 (0.013, 0.819) | |

[a] Abbreviations ref. cat. reference category, CI Confidence interval.

"good" physical health as compared to "poor/fair" (OR 0.316 [95% CI: 0.099, 1.007], p = 0.0515) when they had persistent muscle pain at day 35.

Self-rated general health was not significantly associated with the need for supplemental oxygen at discharge, however, there was a trend indicating an association between requiring supplemental oxygen with a 58% reduction in reporting "very good/excellent" physical health (0R 0.417 [95% CI: 0.167, 1.038], p = 0.0600). In addition, as compared to participants who were discharged without requiring supplemental oxygen, those who required oxygen were 77% less likely to rate their ability to perform daily physical activities as "mostly/completely" 35 days after discharge (OR 0.234 [95% CI: 0.089, 0.617], p = 0.0033), and were 3.2 times more likely to report "some/much" difficulty with walking fast (OR 3.191 [95% CI: 1.321, 7.706], p = 0.0099) or lifting and carrying (OR 2.345 [95% CI: 1.013, 5.425], p = 0.0465). Compared to participants who were discharged without supplemental oxygen, those who were discharged requiring supplemental oxygen were trending towards being significantly (2.2 times) more likely to report "some/more difficulty" with stairs (OR 2.235 [95% CI: 0.979, 5.101], p = 0.0562).

## Discussion

This is the first study of persistence of post-acute symptoms associated with COVID-19 infections in an early US pandemic community-based cohort following hospitalization, and the relationship of symptom persistence to quality of life, health and activities of daily living. The demographics, co-morbid conditions, and percentage of patients who underwent intubation and mechanical ventilation was comparable to those reported by a regionally during the same time [4], however obesity was more prevalent in the cohort.

Although the majority of our participants, 91.9% of men and 81.1% of women, had COVID-19 infections of mild to moderate severity based upon the WHO criteria [14], women had more severe disease and a statistically significant higher odds of experiencing persistent symptoms, demonstrated here to have a statistically significant impact on quality of life estimates and ability to perform activities of daily living. In contrast to our results, COVID-19 infection has been found to have equal prevalence in men and women, with a greater severity in men [21, 22]. In the US local pandemic, men had a higher prevalence of hospitalization [4], perhaps related to behavior, exposure, or intrinsic risk. The literature suggests that women were more likely to experience and more frequently report somatic symptoms in both medical and community settings, attributed to physiologic as well as socialization factors [23]. A study of 1,291 patients with moderate to severe exacerbations of asthma found that women were more likely to report severe complaints of symptom frequency, intensity, and resulting limitations in activity, particularly with moderate exacerbations [24]. It is imperative that the reports of significant symptoms by women not be discounted and this report of greater severity of illness be evaluated and substantiated by other investigation.

Our study found the most prevalent and persistent symptoms at 35 days were fatigue (55.0%; 41% rated moderate, severe, or very severe), and dyspnea (45.3%), accompanied by some or much difficulty with walking (15.6%), lifting and carrying (25.5%), walking up stairs (29.9%), and walking fast (45.6%), and that the persistence of symptoms has an important impact on general, physical and mental health status, social functioning and quality of life within 35 days of discharge, when further evaluation and intervention should be initiated.

The Italian study [11] of persistence of symptoms and quality of life after initial symptoms of COVID-19 infection was undertaken at 60 days and was likely due to the same viral species [25]. These investigators also found fatigue, dyspnea, joint pain and chest pain to be the most prevalent symptoms, indicating the need for earlier assessment and evaluation. The mean age

reported was the same, with a length of stay twice that seen in our study; it is unknown if their cohort included a different racial and ethnic composition (not reported).

Although COVID-19 has clinically distinct manifestations, it is notable that associations between dyspnea and fatigue and the pulmonary physiologic impact of these symptoms were also reported in the severe acute respiratory syndrome (SARS) and the Middle East respiratory syndrome (MERS) coronavirus infections. A meta-analysis of clinical outcomes of SARS and MERS at two months after discharge revealed impaired diffusing capacity for carbon monoxide (DLCO) in 27% and reduced exercise capacity on a six-minute walking distance (6MWD) [26]. Another study of SARS survivors at two years, 27% of whom were health care workers, found that lung function, specifically forced expiratory volume in one minute (FEV1), forced vital capacity (FVC) and total lung capacity (TLC), were reduced in 11–18%. DLCO was < 80% of predicted values for 53%. Their 6MWD was less than the general population but increased significantly to its ceiling at six months. Most importantly, 37% had not been able to return to work at two years [27]. Based on these data, early recognition of dyspnea and functional limitations after resolution of the acute COVID-19 infection requires ongoing evaluation. An Italian study of inpatient acute rehabilitation found that a post-acute care unit dedicated to COVID-19 patients required twice the number of staff and amount of instrumental equipment as compared to two nonCOVID-19 units, resulting in significant excess costs [28].

Our study also revealed that 16.9% of participants rated their mental health as poor or fair. Post-traumatic stress disorder (39%), depression (33%), and anxiety (30%) has been reported more than six months after discharge for MERS and SARS survivors [26]. SARS has been associated with chronic fatigue (40.3%) and active psychiatric illnesses (>40%) persisting to 41.3 months [29]. These data also indicate the need for early mental health assessment and intervention in COVID-19 survivors.

Overall, no significant associations were found between pre-existing conditions (comorbidities at admission) and post-COVID evaluations of health and quality of life as captured through mental health and social relationships. This could suggest that any reports of poor health and quality of life post-COVID could be attributed to patients' COVID illness that might have not been completely resolved given the number of persistent symptoms 35 days post-discharge.

At the time of hospital discharge, 52% of the 183 participants were employed. By day 35, only 49 (29.9%) had returned to work by day 35; 84 (73.7%) who had not, also reported persistent symptoms. The influence of persistent symptoms on the ability to return to work requires further investigation.

Symptoms prevalent at hospital discharge for COVID-19 infection often persist to 35 days, impair individuals' ability to perform daily living activities and quality of life, health, mental, social, and physical function. Identification of symptoms requiring early intervention is critical to planning for and providing post-acute medical, psychological, and physical services to enable recovery from COVID-19 infection, including the ability to return to work.

## Supporting information

**S1 File.**
(PDF)

## Author Contributions

**Conceptualization:** Laurie G. Jacobs, Elli Gourna Paleoudis, Marygrace Zetkulic, Judy L. Aschner.

**Data curation:** Laurie G. Jacobs, Elli Gourna Paleoudis, Dineen Lesky-Di Bari, Anjali Gupta, Lily Rasouli, Bindu Balani, Chinwe Ogedegbe, Harinder Bawa, Lauren Berrol, Nabiha Qureshi.

**Formal analysis:** Elli Gourna Paleoudis, Themba Nyirenda.

**Methodology:** Laurie G. Jacobs, Elli Gourna Paleoudis.

**Project administration:** Laurie G. Jacobs, Marygrace Zetkulic.

**Supervision:** Laurie G. Jacobs.

**Writing – original draft:** Laurie G. Jacobs, Elli Gourna Paleoudis, Themba Nyirenda, Tamara Friedman.

**Writing – review & editing:** Laurie G. Jacobs, Elli Gourna Paleoudis, Themba Nyirenda, Tamara Friedman, Judy L. Aschner.

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
