## [Decision Letter · Decision Letter 0]

28 Sep 2020

PONE-D-20-25200

Persistence of Symptoms and Quality of Life at 35 Days After Hospitalization for COVID-19 Infection

PLOS ONE

Dear Dr. Jacobs,

Thank you for submitting your manuscript to PLOS ONE. After careful consideration, we feel that it has merit but does not fully meet PLOS ONE’s publication criteria as it currently stands. Therefore, we invite you to submit a revised version of the manuscript that addresses the points raised during the review process.

We look forward to receiving your revised manuscript.

Kind regards,

Giordano Madeddu

Academic Editor

PLOS ONE

Journal Requirements:

"The authors disclose the following financial relationships present during the 36 months prior to publication, which did not present any conflicts of interest:

Laurie G. Jacobs -no conflicts of interest, no financial relationships

Elli Gourna Paleoudis -no conflicts of interest, no financial relationships

Dineen Lesky-DiBari -    no conflicts of interest, no financial relationships

Anjali Gupta -no conflicts of interest, no financial relationships

Lily Rasoulli - no conflicts of interest, no financial relationships

Themba Nyrirenda - no conflicts of interest, no financial relationships

Tamara Friedman - no conflicts of interest, no financial relationships

Marygrace Zetkulic -no conflicts of interest, no financial relationships

Bindu Balani - Financial relationships:             Gilead- PI, Remdisivir clinical trial, 2020, personal fees; Merck advisory committee; Abbvie speaker, personal fees

Chinwe Ogedegbe -no conflicts of interest, no financial relationships

Harinder Bawa -no conflicts of interest, no financial relationships

Nabiha Qureshi -no conflicts of interest, no financial relationships

Lauren Berrol -no conflicts of interest, no financial relationships

Judy Aschner -Gilead Science, stock owner"

Reviewers' comments:

Reviewer's Responses to Questions

**Comments to the Author**

1. Is the manuscript technically sound, and do the data support the conclusions?

Reviewer #1: Partly

Reviewer #2: Yes

2. Has the statistical analysis been performed appropriately and rigorously? 

Reviewer #1: No

Reviewer #2: Yes

3. Have the authors made all data underlying the findings in their manuscript fully available?

Reviewer #1: No

Reviewer #2: Yes

4. Is the manuscript presented in an intelligible fashion and written in standard English?

Reviewer #1: Yes

Reviewer #2: Yes

5. Review Comments to the Author

Reviewer #1: The article by Laurie G. Jacobs et al. describes the characterization of the prevalence and persistence of symptoms of COVID-19 following hospital discharge and their impact in several parameters, such as quality of life, mental, physical and psychosocial function.

The topic has been already studied in the literature, and the quantity missing data make this work unattractive. In particular, it seems to be important to underline the lacking of data related to previous patients’ conditions in order to understand the real impact of SARS-CoV-2 infection on investigated parameters. Although previous co-morbidities are present in the baseline description, no data are shown regarding the quality of life and the social, psychological and physical conditions of enrolled patients.

Moreover, the description of the prevalence and persistence should be improved, clarifying aspects within the text and placing into the tables the number of patients with persistence of each symptom during follow-up. The percentage values should be reported in the denominator. In the manuscript it is unclear.

On line 219, the affirmation “…95.4% of all participants […] were classified as having a mild severity of illness requiring no oxygen…” seems not to be in agreement with the previous sentence: “The majority of participants […] required oxygen support…” (line 218).

The OR value and relative CI on line 49 of the abstract and in table 4 (third row) seem to be inaccurate (OR 0.092, CI 0.026-0.033).

Reviewer #2: Jacobs et al present an interesting manuscript about medium-term consequences/persistence of COVID-19 symptoms. This is the first time that a paper explore follow-up experiences with COVID-19, even though only by phone or email interview, and an important point of view over this new disease.

Authors show through sound results and an excellent statistical analysis that COVID-19 symptoms may persist long after the viral RNA is not detectable anymore. Moreover, they also show that older patients, and in particular female ones, have significant higher odds of experiencing persistent symptoms.

The manuscript is well organized and clearly written.

6. PLOS authors have the option to publish the peer review history of their article (what does this mean?). If published, this will include your full peer review and any attached files.

Reviewer #1: No

Reviewer #2: No

---

## [Author Response · Author response to Decision Letter 0]

4 Nov 2020

Responses to Reviewer’s comments:

Reviewer #1: 

The article by Laurie G. Jacobs et al. describes the characterization of the prevalence and persistence of symptoms of COVID-19 following hospital discharge and their impact in several parameters, such as quality of life, mental, physical and psychosocial function. The topic has been already studied in the literature, and the quantity missing data make this work unattractive. In particular, it seems to be important to underline the lacking of data related to previous patients’ conditions in order to understand the real impact of SARS-CoV-2 infection on investigated parameters. Although previous co-morbidities are present in the baseline description, no data are shown regarding the quality of life and the social, psychological and physical conditions of enrolled patients. 

Moreover, the description of the prevalence and persistence should be improved, clarifying aspects within the text and placing into the tables the number of patients with persistence of each symptom during follow-up. The percentage values should be reported in the denominator. In the manuscript it is unclear.

On line 219, the affirmation “…95.4% of all participants […] were classified as having a mild severity of illness requiring no oxygen…” seems not to be in agreement with the previous sentence: “The majority of participants […] required oxygen support…” (line 218).

The OR value and relative CI on line 49 of the abstract and in table 4 (third row) seem to be inaccurate (OR 0.092, CI 0.026-0.033).

Author’s Response:

There is a literature regarding quality of life, mental, physical and psychosocial function but only one prior to this submission, as referenced in the manuscript (refence #11), which assesses of quality of life, mental, physical and psychosocial function with regard to patients surviving COVID-19. The one published prior to the submission of this manuscript, from Italy, cited, represents a more limited assessment using one question, not a standardized tool, and did not have prior assessments of quality of life. The lack of baseline quality of life assessments prior to the patients’ contracting COVID-19 and prior to admission to hospital is not optimal, however, it was not possible to identify community patients who would be admitted, and upon admission, these data could not be collected due to the patient’s severity of illness. 

We did, however, include one “baseline” question about patients’ evaluation of their pre-COVID-19 status regarding general health. We have primarily focused on the post-hospital course, partially to avoid recollection bias since we did not have the option of talking to these patients prior to or at admission. A study design that anticipated admission would have been ideal but not feasible during the pandemic surge period of the study unfortunately. To respond to reviewer #1, we have added text in the results section shown below on new lines 250-255, a new Table 4, also below, which presents the associations between the patients’ response to the question regarding their pre-COVID-19 general health and the general health and quality of life at the time of the survey, 35 days later. Patients were asked about these parameters referencing their COVID-19 illness, using the PROMIS instruments to provide a standard tool and response. In addition, comments were added to the discussion section

Lines 250-255:

Comparing patients’ baseline general health (pre-COVID-19 illness) to post-COVID-19 general health (at time of survey) indicated that patients who reported excellent/very good or good health post-COVID-19 were more likely to have reported an excellent/very good or good pre-COVID health status (Table 4). Similarly, reporting an excellent/very good quality of life, physical health and mental health post-COVID-19 was more frequent among those reporting excellent/very good pre-COVID health. 

Table 4. Associations Between Pre-COVID-19 General Health and Post-COVID-19 (35 Days) General Health and Quality Of Life 

Outcome Effect 

(reference. category: Poor, Fair) Odds Ratio 95% CI P-Value

General Health

Very Good, excellent pre-COVID-19 Good 5.248 (1.093,25.20) 0.0384

General Health

Good pre-COVID-19 Good 4.017 (1.130,14.28) 0.0317

General Health

Very Good, excellent pre-COVID-19 Very good, excellent 8.847 (2.178,35.93) 0.0023

General Health

Good pre-COVID-19 Very good, excellent 2.442 (0.792,7.533) 0.1203

Quality of Life

Very Good, excellent pre-COVID-19 Good 2.916 (0.618,13.75) 0.1762

Quality of Life 

Good pre-COVID-19 Good 1.296 (0.383,4.390) 0.6767

Quality of Life

Very Good, excellent pre-COVID-19 Very good, excellent 5.478 (1.301,23.07) 0.0204

Quality of Life

Good pre-COVID-19 Very good, excellent 1.285 (0.421,3.920) 0.6594

Physical Health

Very Good, excellent pre-COVID-19 Good 6.500 (1.199,35.23) 0.0300

Physical Health

Good pre-COVID-19 Good 5.500 (1.451,20.85) 0.0122

Physical Health

Very Good, excellent pre-COVID-19 Very good, excellent 13.20 (2.747,63.44) 0.0013

Physical Health 

Good pre-COVID-19 Very good, excellent 4.320 (1.249,14.95) 0.0209

Social Relationship pre-COVID-19 Good 1.587 (0.487,5.173) 0.4439

Social Relationship pre-COVID-19 Very good, excellent 2.121 (0.717,6.279) 0.1744

We have edited the former Table 2 (now 3) as suggested to insert the number of patients with persistence of each symptom. In addition, the denominators are added to data in lines 227-234. 

Line 219 regarding oxygen use and the WHO classification has been clarified in the text as: “The majority of participants (89.6% overall; 87.4% of males, 92.7% of females) required oxygen support during hospitalization (Table 2). The WHO severity criteria[14], identified that 95.4% of all participants, 91.9% of males, and 81.1% of females, were classified as having a mild severity of illness, which includes those requiring no oxygen, or oxygen by nasal prongs or nonrebreather mask.” As well, an additional table, now Table 2, below, has been added to further clarify this point.

Table 2. Oxygen Requirements During Hospitalization for COVID-19

 All Participants Male Sexa Female Sexa WHO Severity15

Oxygen Delivery Requirement Count (% of 183) Count (% of 111) Count (% of 69) 

None 19 (10·4) 14 (12·6) 5 (7·2) Mild

Nasal prongs 118 (72·4) 73 (65·8) 43 (62·3) Mild

Nonrebreather mask 23 (12·6) 15 (13·5) 8 (11·6) Mild

High-Flow 15 (9·2)) 6 (5·4) 8 (11·6) Severe

Intubation and mechanical ventilation 8 (4·94) 3 (2·7) 5 (7·2) Severe

The OR and Relative CI on line 49 of the abstract and now in Table 5 have been corrected as “odds rating general health (41.5%, OR 0.093 [95% CI: 0.026, 0.329], p=0.0002),

Tables 2, 3, 4 have been relabeled as Tables 3, 5, and 6 as a two new Tables, Table 2 and Table 4 have been added.

With regard to the issue of the impact of co-morbid conditions present on admission, and their impact on the quality of life and the social, psychological and physical conditions of enrolled patients, we have added the following text on lines 273-285, Table 6, and additional text on lines 384 – 389 in the Discussion Section below:

The most prevalent comorbidities - hypertension, obesity, diabetes and cardiovascular disease - did not appear to have an effect on post-COVID-19 quality of life responses (captured at the time of the survey and during the prior seven days). Similarly, these co-morbidities did not effect post-COVID-19 mental health, social relationships and general health responses. Prior comorbidities, however, did affect post-COVID-19 physical health responses. Patients with hypertension and diabetes, in particular, were less likely to report good or very good/excellent physical health. Those with hypertension and good physical health responses trended to be less likely (OR 0.428, [95% CI: 0.199,0.921], p-value= 0.0300), and physical health responses in diabetics of good (OR 0.330, [95% CI: 0.145,0.747], p-value= 0.0079) and very good/excellent (OR 0.308, [95% CI: 0.138,0.686], p-value= 0.0040 ). In addition, responses of excellent, very good and good physical health were more likely to be reported among those who reported having no known comorbidities at admission (OR= 2.822, p-value: 0.05 and OR= 3.600, p-value 0.01 for excellent/very good health and good health respectively).

Table 6. Associations between quality of life outcomes and persistent symptoms of COVID-19 patients at day 35

Quality of life outcome variable Outcome comparison category Odds ratio (95% CI) P-value

General health 

(ref. cat.: poor, fair)a good 0.530 (0.136, 2.057) 0.3587

 very good, excellent 0.093 (0.026, 0.329) 0.0002

Quality of life

(ref. cat.: poor, fair) good 0.536 (0.159, 1.809) 0.3149

 very good, excellent 0.116 (0.038, 0.364) 0.0002

Physical health

(ref. cat.: poor, fair) good 0.384 (0.098, 1.504) 0.1694

 very good, excellent 0.055 (0.016, 0.193) <0.0001

Mental health

(ref. cat.: poor, fair) good 0.286 (0.061, 1.342) 0.1125

 0.093 (0.021, 0.418) 0.0019

Social relationships

(ref. cat.: poor, fair) very good, excellent 0.245 (0.123, 0.486) 

Social active role

(ref. cat.: poor, fair) good 0.196 (0.060, 0.637) 

 very good, excellent 0.095 (0.031, 0.291) 

Daily physical activity

(ref. cat.: a little not at all) moderately 0.584 (0.058, 5.925) 

 mostly, completely 0.059 (0.008, 0.451) 

Emotional

(ref. cat: always, often) sometimes 0.876 (0.252, 3.051) 

 rarely, never 0.301 (0.096, 0.947) 

Fatigue

(ref. cat.: severe, very severe) moderately 0.786 (0.085, 7.276) 

 none, mild 0.104 (0.013, 0.819) 

 Table 6 Legend

aAbbreviations ref. cat. reference category, CI Confidence interval

Lines 384-389

Overall, no significant associations were found between pre-existing conditions (comorbidities at admission) and post-COVID evaluations of health and quality of life as captured through mental health and social relationships. This could suggest that any reports of poor health and quality of life post-COVID could be attributed to patients’ COVID illness that might have not been completely resolved given the number of persistent symptoms 35 days post-discharge.

---

## [Decision Letter · Decision Letter 1]

1 Dec 2020

Persistence of Symptoms and Quality of Life at 35 Days After Hospitalization for COVID-19 Infection

PONE-D-20-25200R1

Dear Dr. Jacobs,

We’re pleased to inform you that your manuscript has been judged scientifically suitable for publication and will be formally accepted for publication once it meets all outstanding technical requirements.

Kind regards,

Giordano Madeddu

Academic Editor

PLOS ONE

Additional Editor Comments (optional):

Reviewers' comments:

Reviewer's Responses to Questions

**Comments to the Author**

1. If the authors have adequately addressed your comments raised in a previous round of review and you feel that this manuscript is now acceptable for publication, you may indicate that here to bypass the “Comments to the Author” section, enter your conflict of interest statement in the “Confidential to Editor” section, and submit your "Accept" recommendation.

Reviewer #1: (No Response)

Reviewer #2: All comments have been addressed

2. Is the manuscript technically sound, and do the data support the conclusions?

Reviewer #1: Yes

Reviewer #2: Yes

3. Has the statistical analysis been performed appropriately and rigorously? 

Reviewer #1: Yes

Reviewer #2: Yes

4. Have the authors made all data underlying the findings in their manuscript fully available?

Reviewer #1: Yes

Reviewer #2: Yes

5. Is the manuscript presented in an intelligible fashion and written in standard English?

Reviewer #1: Yes

Reviewer #2: Yes

6. Review Comments to the Author

Reviewer #1: I read the revised version of the manuscript.

The Authors added several information and clarifications in the work, making it more attractive.

Reviewer #2: The authors revised the manuscript according to the previous observations made by the reviewers.

The article was improved and it is now acceptable for publication.

7. PLOS authors have the option to publish the peer review history of their article (what does this mean?). If published, this will include your full peer review and any attached files.

Reviewer #1: No

Reviewer #2: No

---

## [Editor Report · Acceptance letter]

3 Dec 2020

PONE-D-20-25200R1 

Persistence of symptoms and quality of life at 35 days after hospitalization for COVID-19 infection 

Dear Dr. Jacobs:

I'm pleased to inform you that your manuscript has been deemed suitable for publication in PLOS ONE. Congratulations! Your manuscript is now with our production department. 

Kind regards, 

on behalf of

Dr. Giordano Madeddu 

Academic Editor

PLOS ONE